# A Smartphone-Based Whole-Cell Array Sensor for Detection of Antibiotics in Milk

**DOI:** 10.3390/s19183882

**Published:** 2019-09-09

**Authors:** Mei-Yi Lu, Wei-Chen Kao, Shimshon Belkin, Ji-Yen Cheng

**Affiliations:** 1Research Center for Applied Sciences, Academia Sinica, Taipei 11529, Taiwan; 2Department of Plant and Environmental Sciences, The Alexander Silberman Institute of Life Sciences, The Hebrew University of Jerusalem, Jerusalem 91904, Israel; 3Department of Mechanical and Mechatronic Engineering, National Taiwan Ocean University, Keelung 20224, Taiwan; 4Institute of Biophotonics, National Yang-Ming University, Taipei 11221, Taiwan; 5College of Engineering, Chang Gung University, Taoyuan 33302, Taiwan

**Keywords:** antibiotics, smartphone, food, biosensors, bioluminescence, milk, whole-cell biosensor

## Abstract

We present an integral smartphone-based whole-cell biosensor, LumiCellSense (LCS), which incorporates a 16-well biochip with an oxygen permeable coating, harboring bioluminescent *Escherichia coli* bioreporter cells, a macro lens, a lens barrel, a metal heater tray, and a temperature controller, enclosed in a light-impermeable case. The luminescence emitted by the bioreporter cells in response to the presence of the target chemicals is imaged by the phone’s camera, and a dedicated phone-embedded application, LCS_Logger, is employed to calculate photon emission intensity and plot it in real time on the device’s screen. An alert is automatically given when light intensity increases above the baseline, indicating the presence of the target. We demonstrate the efficacy of this system by the detection of residues of an antibiotic, ciprofloxacin (CIP), in whole milk, with a detection threshold of 7.2 ng/mL. This value is below the allowed maximum as defined by European Union regulations.

## 1. Introduction

The global use of antibiotic compounds in animal husbandry, partly for disease treatment but mostly for disease prevention and “growth promotion”, is constantly on the increase. The amount of antibiotics globally consumed by livestock has already surpassed that used by humans [1,2], and is expected to increase to over 100,000 tons per year by 2030 [2,3]. Consequently, the growing risk of human exposure to antibiotics in food, particularly meat and other animal-derived products such as milk and eggs, has led regulatory authorities worldwide to pose restrictions on antibiotics residues in food (e.g., European Union (EU) Commission Regulation no 37/2010) [4]. Meeting the requirements of these regulations necessitates accurate and sensitive analytical quantification methods, such as liquid chromatography tandem mass spectrometry (LC/MS/MS). Detection thresholds of ca. 0.5 ng/g in milk products [5] and 0.13 ng/mL in milk powder [6] have been reported using this technique. 

An alternative approach for the detection of trace antimicrobials is the use of microbial whole-cell biosensors, genetically engineered to generate a quantifiable signal in the presence of either specific antibiotics [7,8] or antibiotic groups sharing a mode of action [9,10]. Such genetically engineered bacterial bioreporters have been developed, in which genes encoding a reporter entity (often bacterial bioluminescence genes, *luxCDABE*) were fused downstream of an antibiotic-induced gene promoter [7,8,10]. As the concentrations at which such compounds inhibit bacterial activity are often higher than the required detection thresholds for antibiotic materials in food, several molecular approaches were employed to enhance the bacterial bioreporters’ response to antibiotics and reduce detection thresholds to practical levels [11].

In order to turn a bacterial bioreporter strain into a biosensor device, it needs to be integrated into a hardware platform that will not only house the live cells but will also serve to transduce their biological signals into a quantitative response [12]. We have recently described a portable platform, LumiSense, for such purposes, based on 16 individual bioluminescent wells of a whole-cell sensor array chip with a windowless linear charge-coupled device (CCD) [13,14]. Using ciprofloxacin (CIP) as a model antibiotic, we have demonstrated response times of between 20 and 80 min, and detection thresholds of 8 ng/mL for milk, egg white, and chicken extract, and 64 ng/mL for egg yolk. These values are below the maximal allowed CIP concentrations (100 ng/g) according to EU regulations [4].

In the present study, we report the successful adaptation of this testing strategy to a more advanced hardware environment, based on a combination of a microfluidic system for sample acquisition and a mobile phone for low-light bioluminescence imaging and calculation of the results. 

Due to their advanced optical capabilities, software adaptability, and extreme flexibility with regard to interfacing diverse applications and add-ons, smartphones have recently become a very popular chassis for numerous sensor and biosensor systems. In this respect, the present study is part of a large body of recent publications describing the potential of smartphone-based sensor systems for rapid, laboratory-independent analysis of a broad range of possible targets [15,16,17,18,19,20,21,22,23,24,25,26]. Many of the phone-based sensor systems described in the scientific literature are oriented toward medical applications, with a major focus on point-of-care diagnostics. The diverse biological signals detected, analyzed, and displayed by such systems include optical (chemiluminescence, bioluminescence, fluorescence, color, reflectance) or electrical (amperometric, potentiometric, and impedimetric) outputs [15,23,24,25,26]. Similarly diverse are the biological entities involved in target recognition and signal generation—enzymes, antibodies, nucleic acids, and—as is the case here—live cells. The latter category includes cancer cell cultures, white or red blood cells, embryonic kidney cells [15,23,24], and at least one report of the use of a bioluminescent yeast strain [25]. A smartphone-based biosensor for the detection of bacteria-generated bioluminescence has also been described, employing a relatively large sample tube and off-phone data analysis [26]. 

In the report submitted herein, we have immobilized genetically engineered bioluminescent *Escherichia coli* bioreporter cells in individual microwells of a specially designed microfluidic chip attached to the mobile phone. Furthermore, we have embedded the necessary software, the LCS_Logger applet, in the phone itself, and have employed the built-in optical filter and the computing power of the smartphone to separate image colors and to enhance the signal-to-noise (S/N) ratio. The efficacy of this LumiCellSense (LCS) system is demonstrated by the detection of trace concentrations of an antibiotic, CIP, in milk. In comparison with our previous LumiSense system [14], the LCS retains the sensitivity of the earlier device, while presenting a significantly smaller size and enhanced operational convenience. Sample volumes are reduced (5 µL as compared to 20 µL), the microreactor is smaller, and energy is provided by a battery instead of by benchtop AC (alternating current) outlet. Automatic image capturing and analysis is carried out by a commercial CMOS (complementary metal-oxide-semiconductor) camera with a built-in applet rather than by a home-developed CCD camera and an external computer.

## 2. Experimental Setup

### 2.1. LCS System Design

The LCS system is composed of a smartphone (iPhone SE, Apple Inc. Cupertino, CA, USA) and a miniaturized bioreactor, enclosed in a light-impermeable case (size: 90 mm × 105 mm × 150 mm) (Figure 1a). The bioreactor (Figure 1b) consists of a macro lens, a lens barrel, a metal heater tray, and a temperature controller. The lens barrel is used to align the macro lens to the bacterial chip (BacChip). The barrel is made of black plastic to block ambient light and to reduce heat loss from the heater tray. During measurement, the heater tray is used to hold a sealed BacChip, which is composed of the metal BacChip sandwiched between a layer of MicroAmp^TM^ optical adhesive film (Thermo Fisher Scientific, Waltham, MA, USA) and a thin layer of polydimethylsiloxane (PDMS). The BacChip harbors 16 independent wells, 2 mm in diameter and 6 mm in depth, which contain the alginate-immobilized bacterial reporters and the sample. A 12 volt lithium battery with a capacity of 1800 mAh is connected to the temperature controller and the heater tray to power a film heater (Taiwan KLC corporation, Taichung, Taiwan, TSA(C)016d000) so as to maintain the BacChip temperature at 37.1 ± 0.6 °C. Maintenance of optimal temperature is crucial to bacterial bioluminescent activity [13,27,28]. The mass of the heater tray and the BacChip are 17 and 4 g, respectively, ensuring a small heat mass and, hence, rapid rise to target temperature and low power consumption. The maximal transient electrical power consumption was measured to be 2.4 W, ensuring uninterrupted bioreactor operation for 24 h. 

### 2.2. Fabrication of PDMS Layer on BacChip

PDMS is an elastic material with superior oxygen permeability [29]. It is capable of preventing water evaporation while allowing oxygen permeation, so that optimal bacterial activity is maintained. The PDMS layer attached to the BacChip was prepared as follows. A mixture of Sylgard 184 (Dow Corning Corporation) reagents at an A:B ratio of 10:1 was prepared and subjected to vacuum for 20 min to remove bubbles. A layer of the uncured Sylgard 184 mixture was smeared on a clean polymethyl methacrylate (PMMA) base plate, and then covered by the BacChip. A 0.26 mm spacer was inserted between the PMMA plate and the BacChip to fix the uniform thickness of the PDMS layer. Following a 2 h incubation at 85 °C, the PDMS layer adhered firmly to the metal BacChip. The PDMS-coated BacChip was separated from the PMMA base plate and then treated by oxygen plasma (Nordson March AP-600) at 100 W for 20 min to render the surfaces hydrophilic.

### 2.3. Preparation of Bacteria

The bacterial bioreporter employed in the current study was a bioluminescent *E. coli* strain harboring a plasmid-borne fusion of the *E. coli recA* gene promoter to the *Photorhabdus luminescens luxCDABE* gene cassette [30]. This strain has previously been shown to be a highly sensitive sensor of the fluoroquinolone antibiotic CIP [14]. Culture aliquots (35 μL) from a −80 °C stock were transferred to 20 mL lysogeny broth (LB) (BD Difco, Franklin Lakes, NJ, USA) containing 100 µg/mL ampicillin and cultured overnight with 200 rpm agitation at 37 °C. The overnight culture was measured by a M6+ colorimeter (Metertech Inc., Taipei, Taiwan) and then centrifuged at 16,000× *g* for 5 min. The bacteria were resuspended in LB containing 0.4% alginic acid (Sigma Aldrich, St. Louis, MI, USA) to reach cell density of 2.4 × 10^9^ cells/mL. Bacterial/alginate suspension aliquots (5.5 μL, containing 1.32 × 10^7^ cells) were loaded into individual wells, and the alginate was solidified by the addition of 0.5 μL of 2.5% CaCl_2_. After 20 min, the alginate-immobilized bacteria were ready for analysis. The optimized cell density was determined in preliminary experiments in which the response of different bacterial concentrations, ranging from 8 × 10^8^ to 3.2 × 10^9^ colony-forming unit (CFU) per mL (4.4 × 10^6^–1.76 × 10^7^ cells per well) to 16 ng/mL CIP was assayed (Appendix A). 

### 2.4. CIP Preparation and Bioreporter Stimulation

A stock solution (10 mg/L) of ciprofloxacin hydrochloride monohydrate (CIP; Sigma-Aldrich) was prepared in deionized water and kept in the dark to prevent decomposition. The stock solution was diluted in LB or whole milk to final concentrations of 8–128 ng/mL, and then added to each BacChip well already containing the alginate-immobilized bacterial bioreporters, following which the BacChip was sealed by an adhesive film and inserted into the heater tray. A dedicated smartphone application (see Section 2.5) was then employed to initiate the measurements and analyze the results.

### 2.5. Smartphone Application for Image Capture, Processing, and Analysis 

A smartphone program, the “LCS_Logger” applet, was developed on the iOS (Apple Inc.) platform using Swift language (Apple Inc.). The applet automatically acquires time-lapse images of the BacChip at pre-set time intervals and exposure durations. Following image acquisition, images are processed and analyzed by the applet, showing graphic results in real time on the phone screen. As the low luminescence signals emitted by the bacteria are too weak to be observed by a regular smartphone camera, and in view of the limited maximal exposure time (1/3 s) allowed by the phone model used, two major features were included in the LCS_Logger applet. To mimic a prolonged exposure time (~150 s), 450 consecutive images were taken and integrated. Furthermore, the time-lapse images of blue–green 490 nm bioluminescence produced by the activated bacteria, taken in full color, were separated into red, green, and blue channels to enhance the S/N ratio. The accumulated green channel images were then used for signal quantification, relative luminescence intensity (RLI) calculation, and subsequent data analysis. The RLI was defined as:RLI(n, t) = [L(n, t)/L (1,t)]/[L(n,0)/(L(1,0)],(1)
where L(n, t) is the luminescence intensity of the nth well at time t. The first well (n = 1), which contained bacteria without antibiotic stimulation (CIP concentration = 0), served as the control. 

The RLI value is essentially a signal normalized against both the temporal fluctuation of each well and the spontaneous background luminescence from bacteria without antibiotic stimulation. An average and standard deviation (SD) of the RLI of each well was calculated in real time (temporal SD) for the first 10 data points, and was treated as the background luminescence for this well. An RLI increase above this value was considered significant if two consecutive RLI values were higher than the well’s initial average RLI by at least three temporal SDs. The temporal SD represents a single well’s RLI fluctuation over time and is, thus, different from the SD among different wells.

### 2.6. Measurement of Absolute Sensitivity of the LCS System

The light from a LED light source (470 nm, FWHM 20 nm, Thorlabs, Newton, NJ, USA, M470F1) was coupled into an optical fiber (Thorlabs, M71L01), the output of which was collimated by a collimator (Thorlabs, F230SMA-B). The light power out of the collimator was measured by a power meter (Thorlabs, Meter PM100A, Sensor head S120VC) and the illumination area by the collimator was measured. The photon density was then calculated by dividing the photopower by the illumination area. The LED light, attenuated by neutral density (ND) filters (Thorlabs, MDK01) was projected upon an “empty” BacChip; the light passing through the BacChip wells was imaged by the smartphone and quantified in RLI units. Well number 1 was blocked and used as a dark blank, as in Equation (1). The photon flux passing through a single well in the BacChip was obtained by multiplying the light intensity by the area of the well. The photon intensities of the attenuated light were calculated by the attenuation ratio provided by the ND filters. One 3-OD (optical density) filter, two 2-OD filters, two 1-OD filters, and one 0.5-OD filter were used in different combinations to obtain attenuations up to 9.5 OD, and the corresponding RLIs were measured. 

The RLI changes with increasing photon flux were fitted by a rational exponent function, as shown below, to determine the detection limit:RLI (P) = *a* + *b* × (P + *c*)*^d^*,(2)
where P is the photon flux through the well, *a* represents the vertical offset, *b* represents the linear expansion of the curve, *c* represents the horizontal offset, and *d* represents the curvature of the function. The coefficients *a*, *b*, *c*, and *d* were fitted to the experimental data to determine the response curve of the CMOS sensor.

## 3. Results

### 3.1. Photon Sensitivity of LCS System 

To determine the absolute sensitivity of the LCS system, the light of a 470 nm LED light source, modulated by different ND filters to generate a range of light intensities, was shined upon the BacChip, and RLI was calculated by measuring the light passing through a single well. Without attenuation, the light intensity was 4.80 μW/cm^2^ (= 1.13 × 10^14^ photons/s·cm^2^ for photons with a central wavelength of 470 nm) and the corresponding photon flux passing through a single well was calculated to be 3.55 × 10^12^ photons/s (= 1.13 × 10^14^ photons/s·cm^2^ × π × r^2^ cm^2^, where r is the diameter of a well, r = 2 mm). As shown in Figure 2, the RLI increased monotonously but not linearly with increasing photon flux, indicating that the response of the smartphone’s CMOS sensor tends to saturate at the high photon fluxes. While this non-linearity may potentially cause inaccurate quantification of high-intensity signals, this is not the case in the present application, where the bioluminescent signals are characterized by very low intensities. 

The inset in Figure 2 shows the RLI measured at low photon intensities. The response curve in this region was well fitted by Equation (2) with *a*, *b*, *c*, and *d* values of 0.99, 6200, 0.0, and 0.13, respectively, and an error coefficient χ = 9.8 × 10^−8^. Using the average luminescence measured at the dark well plus three times the temporal SD, the detection limit by one well in BacChip was calculated. The photon flux needed to detect a significant RLI change (dashed horizontal line) was calculated to be 12,000 photons/s. In the present experimental setup, this signal was obtained by a 150 s exposure. A longer exposure for signal integration and noise reduction for achieving a lower detection limit is possible, but at the cost of extended test duration.

It should be noted that the LED light source used has a narrower spectral width than that of bacterial bioluminescence. In addition, the CMOS’s built-in filter may exert different blockage ratios for LED-light and bioluminescence. The actual response of the CMOS sensor to the bioluminescence thus needed to be investigated further. For this purpose, live bacteria were introduced to the BacChip wells and the detection performance was investigated as described in the following section.

### 3.2. Stability and Homogeneity of Bioluminescence Detection in the LCS System

The stability and homogeneity of the smartphone’s CMOS sensor are essential for the multiple-well analysis demonstrated in this study. We first examined the noise background (the temporal SD) across the 16 BacChip wells, all containing live bacterial reporters. A typical time-lapse baseline (Figure 3a) exhibited good signal stability for a single dark well, with a SD smaller than 0.002 RLI. The average of the temporal SD among all 15 wells (excluding well number 1, which served as the control) was 0.0016 RLI (Figure 3b), demonstrating signal homogeneity across the different wells. This allows to compare results from separate wells containing different samples, or exposed to different experimental conditions.

### 3.3. Enhancement of Signal Using Color Separation for Image Analysis

Given that the bacterial bioluminescence is a very weak signal and generally requires highly sensitive detection instruments such as photomultiplier tubes (PMTs) and cooled CCDs, the response of the smartphone’s CMOS sensor to bioluminescence needed to be investigated. A comparison of the signals from different color channels was conducted to identify the channel with the highest S/N ratio. The precise location of each well was first pinpointed on a BacChip fitted with phosphorescence tape (Figure 4a), and then time-lapse images of the BacChip containing immobilized bacteria (*recA::luxCDABE*) stimulated with various concentrations of CIP were acquired in full color. The raw white-light signal (Figure 4b) was separated into red (Figure 4c), green (Figure 4d), and blue (data not shown) channels. Bacterial bioluminescence has a spectral maximum of 490 nm, which is close to blue light [28]. As expected, the red channel displayed the lowest signal rise, whereas the green and blue channels both showed significantly larger signals than the white-light signal (Figure 4b–e). Unexpectedly, the green channel showed a slightly higher signal than the blue one, and also displayed an earlier on-time at which the RLI shows a significant increase above the background (Figure 4e). Based on these findings, the green channel data are thus presented for subsequent experiments. It should be noted that since luminescence was measured at a relatively low CIP concentration (32 ng/mL), signal fluctuations were observed. Nevertheless, color-separated channel analysis revealed the different on-time of individual channels: White channel at 51 min, green channel at 42 min, blue channel at 54 min, and red channel at 81 min. A possible explanation for the observation that the blue channel did not yield the expected highest signal is that the spectral properties of the smartphone’s built-in color filter may not fully match the bioluminescence emission spectrum. Figure 4f shows the time course of bioluminescence emitted by the CIP-stimulated bacteria. Using color separation to enhance S/N ratio, bioluminescence stimulated with as low as 8 ng/mL CIP was measured and analyzed. In the screenshot, the blue text and the blue vertical line mark the on-time of each individual well (Figure 4f), allowing the user to obtain real-time analysis results. Testing the same bacterial culture in a commercial photomultiplier-based microtiter plate reader (BioTek Synergy 2) has yielded results that were not dramatically different (Appendix A), in spite of the clearly superior photon detection sensitivity of the photomultiplier. It should be added that the LCS is not only much less bulky than a plate reader, but, in the example in Appendix A, much lower sample volumes were also used (5 µL as compared to 50 µL).

### 3.4. Detection of CIP in Whole Milk

Whole milk spiked with CIP was analyzed by the LCS system. Figure 5a shows the time-lapse luminescence of the reporter bacteria stimulated by three concentrations of CIP for up to 30 h, with the response peak at about 5 h after CIP addition. Figure 5b presents a magnified view of luminescence changes in the first 180 min of this time course. Even at the lowest CIP concentration tested, 8 ng/mL, signal increase above the background was significant at the last hour of the assay (Figure 5b), indicating that the LCS system is capable of detecting low levels of CIP in whole milk. It should be emphasized that although the signal appears to be fluctuating, the observed luminescence is higher than the baseline by over three SDs for at least two consecutive time points. The data discrimination strategy assures the correctness of the analysis. The detected concentration, 8 ng/mL, is close to but lower than the maximum allowed residue limit (10 ng/mL) set by the Taiwan Food and Drug Administration [31], which employs LC/MS/MS to analyze CIP and other animal drugs; it is also lower than the level set by the EU (100 ng/g) [4].

Figure 5c displays the signal at a fixed time point (180 min) as a function of a broader CIP concentration range. To further refine the limit of detection of CIP in the LCS under these experimental conditions, the data in Figure 5c were analyzed by a linear fit to the data points of the lowest concentrations, 8–32 ng/mL, and extrapolated to the value of significant RLI changes. The limit of detection of CIP thus calculated was 7.2 ng/mL.

In our experimental results, the absolute photon flux can be correlated with the photon number emitted by single cells that were induced by CIP. The relatively low number suggests that either not all cells were induced, or that the responses from the cells were not maximal. The low induction may be caused by the fact that each cell interacted with a limited number of CIP molecules. For 10 µL of an 8 ng/mL sample and 1.32 × 10^7^ cells per well, each cell could react with only 10,000 CIP molecules. Assuming the response of the cell to the CIP consumes the molecules, increasing the number of CIP molecules could enhance cell response. In other words, simply by increasing the sample volume at the same concentration, the detection limit might be lowered. 

## 4. Discussion

In the present study, we developed a miniature bioluminescence detection system, LCS, integrating a commercial smartphone (iPhone SE) with a mini-bioreactor for the induction of bacterial bioreporters, which emit light in response to the presence of target analytes. Image acquisition, data processing, and real-time reporting are carried out by an internal custom-built applet, LCS_Logger. The applicability of the LCS system was demonstrated using a bacterial sensor strain harboring a *recA promoter::luxCDABE* plasmid-borne fusion that responds to target compounds by activation of the SOS DNA repair system and are thus not specific to CIP only. Because these bacterial bioreporter cells do not respond to biomolecules such as proteins, lipids, and carbohydrates in the sample matrix, isolation of the target antibiotic from the milk sample matrix is thus not required [14]. The efficacy of this system was demonstrated by the detection of residues of an SOS-inducing antibiotic, CIP, in whole milk, with a detection threshold of 7.2 ng/mL. This value is below the maximal allowed as defined by European Union regulations (100 ng/g). To further demonstrate the flexibility and adaptability of the described system, we have also successfully tested it using different bacterial sensor strains, engineered to detect DNA damaging agents and heavy metals. (Appendix A)

In addition to accurate and sensitive analytical quantification using time-consuming and costly instruments, including LC/MS [5,6] and high-performance liquid chromatography (HPLC) [32], diverse sensors have recently been reported for the determination of CIP in milk. For example, an amperometric magneto immunosensor was reported to measure as low as 0.009 ng/mL CIP [33]; a surface plasmon resonance (SPR) immunosensor [34], and a lateral-flow immunochromatographic assay detected 1.2 ng/mL enrofloxacin, a CIP analog, and 2.5 ng/mL CIP [35], respectively. These highly sensitive and specific methods, based on the interaction of a specific antibody with CIP, either involve complex sample pretreatment (SPR), lengthy detection time (amperometric magneto immunosensor), or lack quantitative information (lateral-flow). Our LCS system is able to detect CIP at the ng/mL level, comparable to both the SPR and lateral-flow immunochromatographic assay. Furthermore, in contrast to these assays, the flexibility of the LCS system allows the determination of other analytes by employing different strains of bacterial sensors (described below and in Appendix A).

Several important features render the LCS system a convenient and sensitive biosensor. Probably the most important of these is that it uses live bacterial cells as bioreporters, which not only report the presence of the target but also provide information on its bioavailability and toxicity. Different physiological factors can reduce the effectiveness of a drug before it enters systemic circulation, including gastrointestinal absorption efficiency, degradation or metabolism before absorption, or the first pass effect of liver [36]. Given that the bioavailability of CIP is indeed decreased by concomitant ingestion of milk or yogurt [37], CIP may be decomposed, metabolized, or interfered by proteins or factors in milk. Thus, the CIP concentration that the LCS “sees” as 8 ng/mL may actually be lower than the initial concentration. As the bacterial bioreporters are “interchangeable”, the specificity/selectivity will depend on the bacterial strain and application.

With the continuous advances in synthetic biology, along with the availability of diverse bioluminescent sensor strains for numerous targets [9], such a device may find additional uses beyond food safety [38], with environmental, agricultural, industrial, or security applications. Similar to the study reported herein, some of these sensors are cell based [15,23,24], but very few of these are based on microorganisms as the sensing entities. This is somewhat surprising in view of the fact that most cell-based sensors described in the scientific literature over the last three decades are microbe-based. Guo et al. describe the use of a smartphone as a fluorescence diagnostic system, employed to image the paper-based gold ions detected in urine by a *Cupriavidus metallidurans* reporter strain [39]. Cevenini et al. described the use of a compact camera, wirelessly connected to a smartphone, for imaging the response of a bioluminescent *Saccharomyces cerevisiae* strain to endocrine disrupting compounds [25]. As far as we are aware, however, the only description of the phone-based quantification of low-intensity microbial luminescence was that of Kim et al. [26]. This report, however, lacks the intricate integration of hardware, software, and microorganism embodied in the LCS. 

Another unique feature of the LCS system is the miniaturized bioreactor that integrates a precise temperature controller, a heater, and an oxygen permeable configuration, which provide an optimal environment for cellular viability and bacterial luciferase activity. The bacterial activity in the LCS system shows a bell-shaped response to CIP stimulation, with maxima approximately after 5 h of continuous operation inside the microreactor. In general, this bell-shaped response might be due to nutrient depletion, drug metabolism, changes in environmental condition (temperature, ionic strength, and pH value), lifetime of the luciferase enzyme, or intrinsic physiological state of the bacterial cells [40,41]. Based on our results, the current setup allowed operation of the LCS system for up to five hours. For longer operation time, a different design that could continuously supply the required nutrition would be necessary. For example, a continuous cell culture system may be useful for such an application [41]. Such a continuous culture system may be further miniaturized by microfluidic technologies.

Furthermore, the multi-well BacChip design allows either simultaneous measurement and comparison of multiple samples or multiple sensor strains responding to several target analytes in a single analysis. 

The LCS system applet provides a simple operation interface that conducts image acquisition and data analysis without external computer processing, enabling data accessibility in real time on the phone screen. The prolonged image integration and color separation in the image analysis enhance the weak bioluminescence signals to be identified. The approach taken by Kim et al. was to use an external computer to run a noise reduction algorithm, which simultaneously lowered the background and enhanced the signal [26]. Significant signals with high S/N ratio from low light were successfully detected in this manner, but all analyses were carried out in a PC environment and not on the smartphone itself.

The LCS system is thus an “all-in-one” device, enabling portable and point-of-care detection. Future challenges that should be overcome in the development of smartphone-based bioluminescent whole-cell biosensors include improving camera sensitivity in low-light detection, upgrading the applet computing algorithm, and extension of the shelf life of the immobilized bacteria. The shelf-life of the bacteria inside the chip is an important property of the whole-cell sensing system. Previously we have used the LumiSense system [14] to test such shelf-life. The result showed that the luminescence intensity from the bacteria in the chips stored at 4 °C decreased to about one half after 48 hours of storage (Appendix A). Extension of the shelf-life will require further investigation.

## Figures and Tables

**Figure 1 sensors-19-03882-f001:**
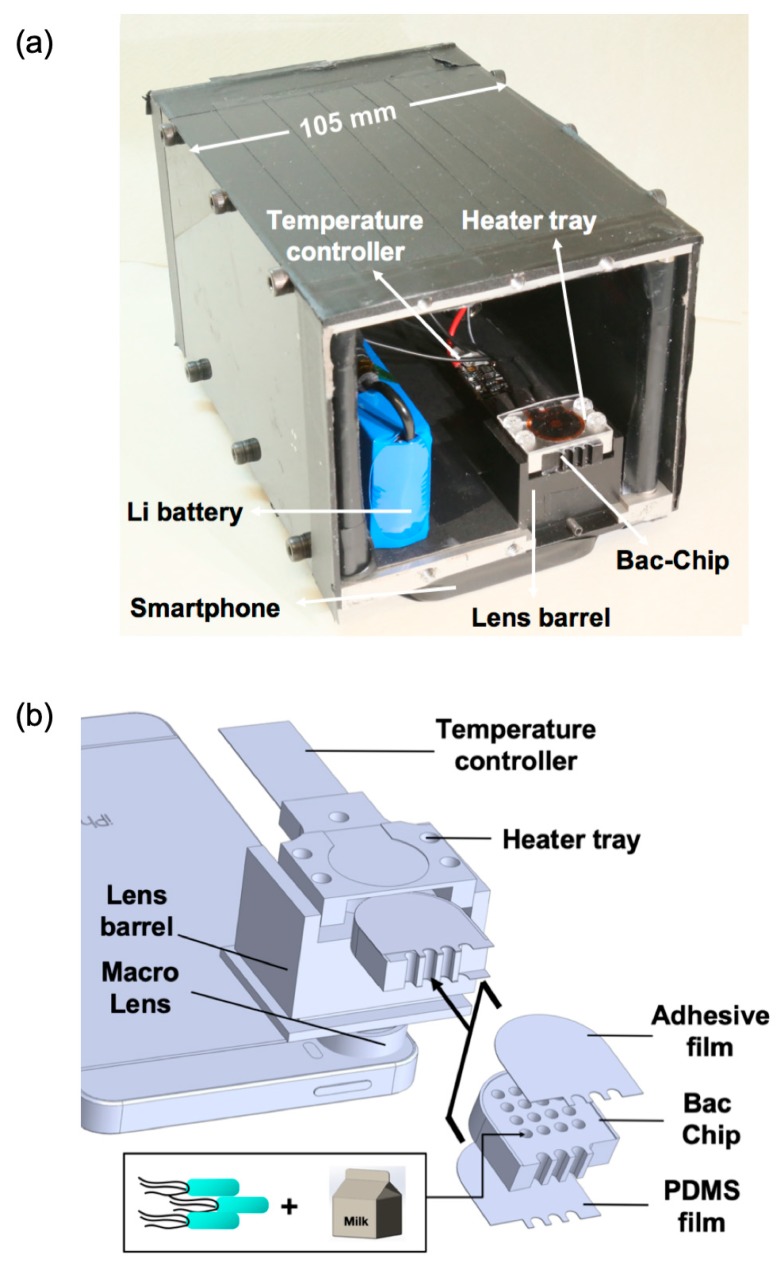
The LumiCellSense (LCS) system. (**a**) Photo of the LCS system. All of the system’s components except the smartphone are enclosed in a chamber for protection from ambient light. (**b**) A schematic diagram of the bioreactor and smartphone. The macro lens is aligned with the smartphone’s camera. The alginate-immobilized bacteria and the sample are loaded into the wells of the bacterial chip (BacChip), which is sandwiched between a polydimethylsiloxane (PDMS) layer and an adhesive film and then inserted into the heater tray.

**Figure 2 sensors-19-03882-f002:**
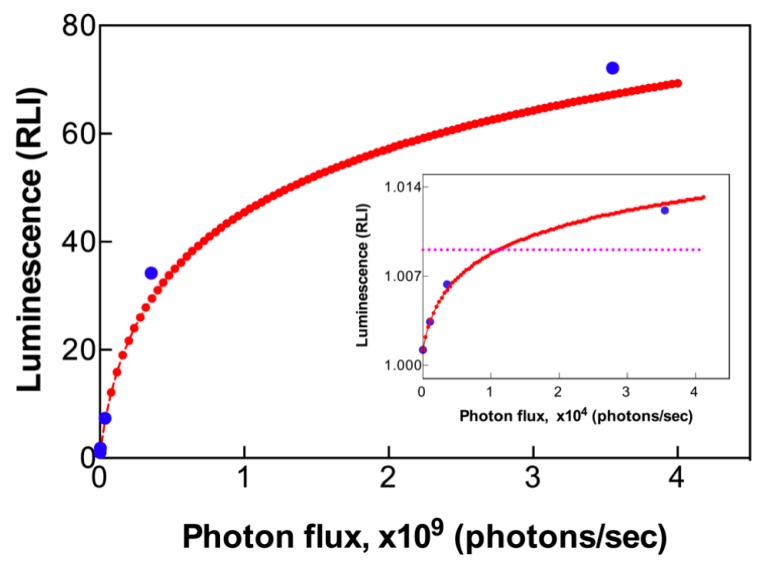
Measurement of absolute photon detection sensitivity. Luminescence intensity was acquired by measuring light from a LED illuminating through a single well of the BacChip, at an exposure time of 150 s. The inset magnifies the data at the low photon intensities range. A rational exponent fitting curve (red dots) was plotted to calculate the detection limit. The dashed horizontal line indicates luminescence intensity that is higher than the background by three standard deviations (SDs) of the background signal intensity.

**Figure 3 sensors-19-03882-f003:**
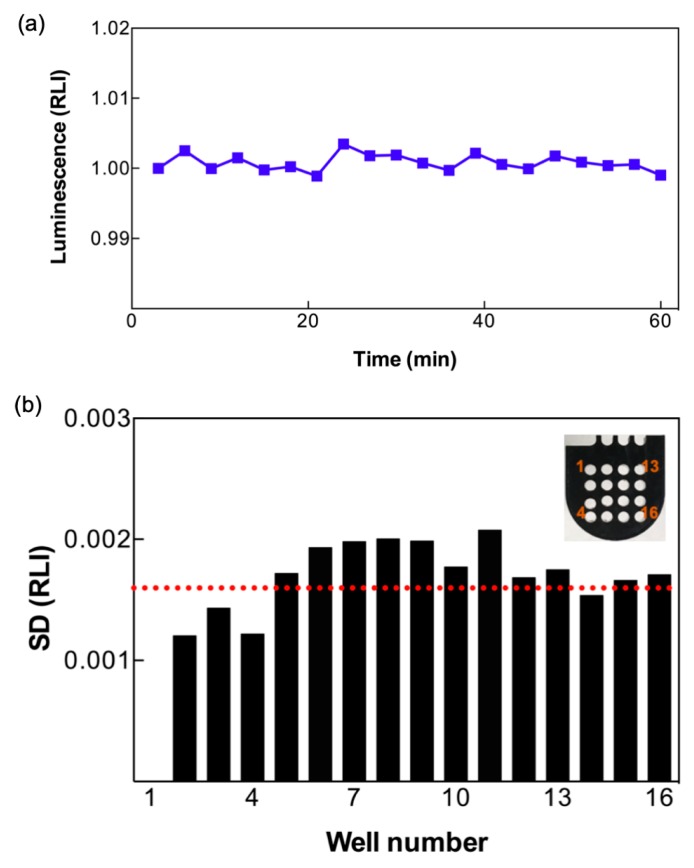
Stability and homogeneity of the LCS system. (**a**) Signal stability of a typical dark well with immobilized bacteria (1.32 × 10^7^ cells per well) without stimulation. (**b**) Temporal SD of relative luminescence intensity (RLI) of different wells in the BacChip. The red dashed line indicates the average SD of 0.0016 RLI. Inset: Close-up photo of a BacChip and the numbering of the wells.

**Figure 4 sensors-19-03882-f004:**
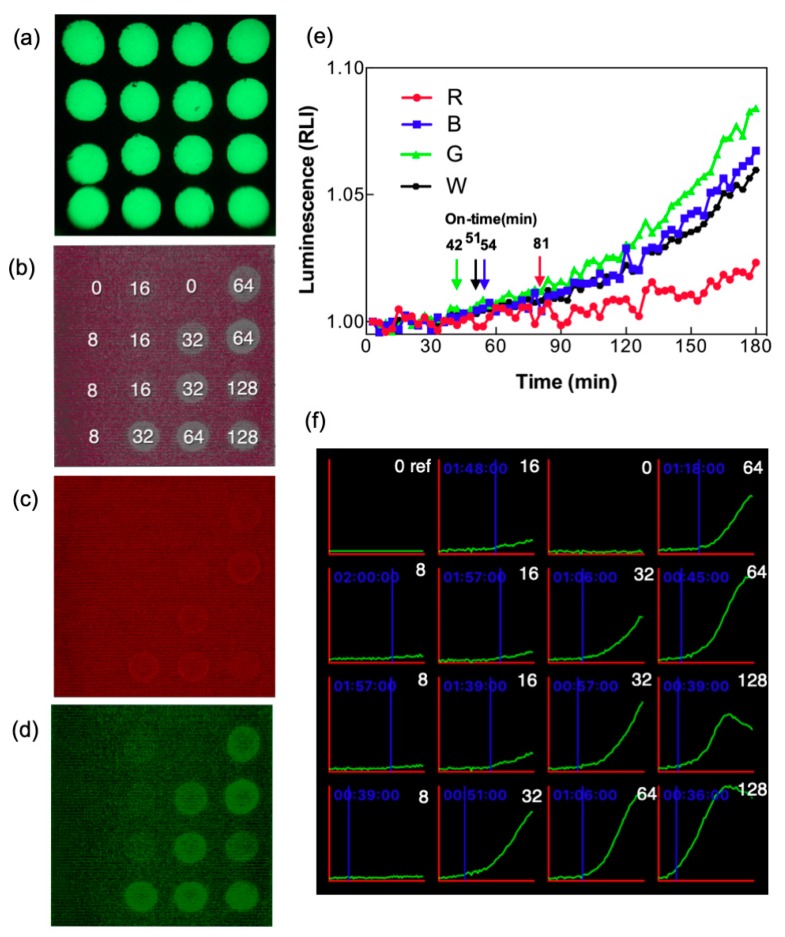
Color separation for image analysis. (**a**–**d**): Smartphone-obtained images showing the (**a**) image of luminescence from phosphor tape for identification of the well positions. (**b**) Image of luminescence from the immobilized bacteria (1.32 × 10^7^ cells per well) stimulated with ciprofloxacin (CIP) for 180 min. The number at each well indicates the CIP concentration in ng/mL. Image (**b**) was isolated into red and green channels, displayed in (**c**) and (**d**), respectively. (**e**) Time-lapse RLI of the white light (W) and of red (R), blue (B), or green (G) channel are from the immobilized bacteria stimulated with 32 ng/mL of CIP. Arrows indicate the on-time calculated for each channel. (**f**) Screenshot of the smartphone showing the time-lapse RLI of each well. The blue lines mark the on-time calculated for each well. The white number indicates the concentration of CIP.

**Figure 5 sensors-19-03882-f005:**
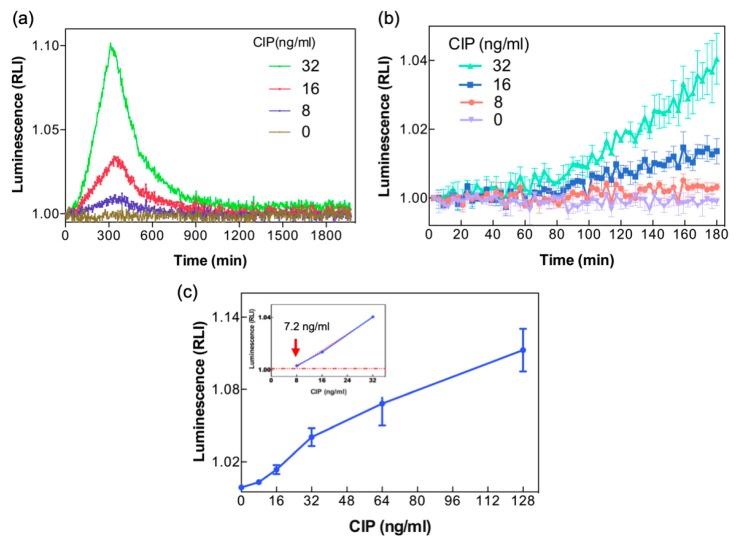
Detection of CIP in whole milk. (**a**) Luminescence of immobilized bacteria (1.32 × 10^7^ cells per well) stimulated with 0, 8, 16, or 32 ng/mL of CIP in whole milk was continuously recorded for up to 30 h. (**b**) Magnified details in the early 180 min of time-lapse luminescence. (**c**) Luminescence at 180 min in response to a broader range of CIP concentrations. The inset shows a linear fit to the data points and extrapolation to determine the detection threshold.

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
