# Peer review of "A Smartphone-Based Whole-Cell Array Sensor for Detection of Antibiotics in Milk"

_sensors, 2019, doi:10.3390/s19183882_

Round 1

Reviewer 1 Report

The article is generally well-written and reports interesting results, some minor points should be addressed to improve the quality.

In the introduction section authors refer to multiplex (“16-multiplex whole cell sensor array”), this could be misleading, multiplexing has not been demonstrated in this work.

“molecularly engineered” should be replaced by “genetically engineered”.

Figure 3 could be placed as supplementary material.

An additional paragraph should be introduced in the discussion section to critically discuss the stability of these living biosensors in the cartrige. Only the stability of the bioluminescent signal is discussed and shelf-life issue is only mentioned in the final line of the manuscript.

Reviewer 2 Report

Dear Editor, Dear Authors,

In manuscript sensors-550739, the authors describe detection of ciprofloxacin with engineered, bioluminescent, “reporter” bacteria. Unfortunately, they did about the same in at least one, already published paper [1]. Thus, this manuscript lacks important elements of novelty. The only element of novelty of the manuscript seems to be related to data acquisition and handling which, in the current form of the sensor, is made entirely on a smartphone (while the previous version of the sensor used a CCD device and a PC). However, bioluminescent, “reporter” cell-based biosensors with data acquisition and processing made entirely on a smartphone have already been reported by others (e.g., in Ref. [2]). How the system described in the manuscript compares to the previously described similar systems? Is it faster? Is it more sensitive? These details are not revealed. Next to this lack of novelty, there are also some other (smaller and larger) problems with the manuscript. Please find them listed below:

1.) There are plenty of other (optical and electrochemical) sensors for the detection of ciprofloxacin which have already been reported in the literature. However, manuscript sensors-550739 does not mention any of them. It does not compare obtained results with the analytical performances of previously published sensors. If the sensor described in the manuscript represents an improvement or not remains a secret in such conditions.

2.) The authors did not compare the obtained results with those facilitated by commercially available instrumentation (or with those facilitated by their previous similar sensor). It would have been nice to see how the developed, smartphone-based system compares with a commercially available system for measuring luminescence. We can expect that the developed system does not equal the performances of a commercial instrument but it would have been important to see how much worse it is.

3.) The wells which host the reporter cells and the samples are described as “specially designed microfluidic chip” in the manuscript. However, those wells (ɸ= 2 mm; H = 6 mm) have no single feature that is specific for “microfluidic chips”. The whole device is described as “hand-held bioluminescence detection system”. However, the device is not that small and requires a three hours long incubation of the samples with the “reporter” cells. Therefore, describing the system as “hand-held” is not the most appropriate. It is not like one can determine the ciprofloxacin content of a milk sample while holding the instrument in her / his hands.

4.) The authors attempt to calibrate the camera of the used smartphone. Therefore, they plot the photon flux produced by a LED through their system vs. the luminescence reported by the smartphone and the developed app. However, this plot seems to be (somehow) nonlinear both at high and at small photon fluxes (see Figure 2 and the inset of Figure 2). The CMOS sensor of the smartphone can be expected to saturate at high photon fluxes but it is not clear why low photon fluxes (i.e. fluxes similar to those produced by bioluminescent bacteria) induce non-linear signals and how this affects the accuracy of the measurements.

5.) It was shown that the bioavailability of ciprofloxacin can be reduced by concomitant ingestion of milk or yogurt [3]. This translates into the fact that bioluminescent, “reporter” cell based ciprofloxacin quantification methods are very likely to underestimate the concentration of ciprofloxacin in milk or yogurt samples. How can this problem be addressed? Some recoveries (determined with spiked samples) would have been nice to present.

6.) Taking into account the noise that characterizes the background signal and the signal due to 8 ng/mL ciprofloxacin (see Figure 5A), I would say that the developed sensor is not able to distinguish in between a milk sample with 8 ng/mL ciprofloxacin and a milk sample without this compound. I would also say that even the signal for 16 ng/mL ciprofloxacin is smaller than 3 times the noise observed in the absence of ciprofloxacin. If we define the detection limit as 3x the noise of the background signal, the detection limit of the developed sensor seems to be around 20 ng/mL (a value that is higher than the maximum residue level imposed in the EU).

7.) Combining enhanced portability (that is assured by using a smartphone as reader) with a 3 hours long analysis procedure does not make too much sense. If I would be a farmer (milk processor) I would skip portability for a slightly bulkier device that provides results in ~ 15 min. Not all sensors are suitable and worth to be combined with reading by a smartphone.

8.) There is no single word about the selectivity of this system. Is it selective for ciprofloxacin only or it will be affected by other similar drugs? There is no single word on the stability and reusability of the sensor. In yet other words, the work is somewhat superficial, based on very few experiments.

Taking into account the lack of important novelty and these 8 smaller and larger problems, I am recommending rejection of the manuscript.

References:

[1] Kao, W.-C.; Belkin, S.; Cheng, J.-Y. Microbial Biosensing of Ciprofloxacin Residues in Food by a Portable Lens-Free CCD-Based Analyzer. Anal. Bioanal. Chem. 2018, 410 (4), 1257–1263. https://doi.org/10.1007/s00216-017-0792-x.

[2] Cevenini, L.; Calabretta, M. M.; Tarantino, G.; Michelini, E.; Roda, A. Smartphone-Interfaced 3D Printed Toxicity Biosensor Integrating Bioluminescent “Sentinel Cells.” Sens. Actuators B Chem. 2016, 225, 249–257. https://doi.org/10.1016/j.snb.2015.11.017.

[3] Neuvonen, P. J.; Kivistö, K. T.; Lehto, P. Interference of Dairy Products with the Absorption of Ciprofloxacin. Clin. Pharmacol. Ther. 1991, 50 (5–1), 498–502. https://doi.org/10.1038/clpt.1991.174.

Reviewer 3 Report

In this paper, the authors present an integral smartphone-based whole-cell biosensor which is aimed at the detection of residues of an antibiotic, ciprofloxacin, in whole milk. Although the paper is of interest to the field, the paper still needs improvement to be appropriate for publication in sensors. The comments and suggestions are detailed below.

It is not very clear to me why this study is better as concerns as the CIP detection than the previous study of the authors (Reference No 14). The authors in the previous study manage to detect CIP faster than this study (20-80min) compare to this study (180min). Please explain better, in the introduction, the reason of the importance of this study.

In the introduction, another point is that authors did not say which are the maximal allowed concentrations by EU regulations. Please add these values.

In 2.3 section, the abbreviation CIP is referred for first time. Please write whole word and put CIP in parenthesis.

In experimental section, authors should write the statistical analysis that they used in order to be reasonable the significant differences that they referred to.

In results, I am not sure how the authors came to the conclusion that the detection limit is 7.2 ng/mL. The range of CIP was 8-32 ng/mL. Did you try less than 8ng/mL?

Also, if we observed the figure 4b, no light appears at CIP concentration 8 ng/mL on contrary there is slight fluorescence at 16 ng/mL. Could you explain this result and how is it possible to detect less than 8ng/mL if no light emission exists even at 8ng/mL?

In section 3.4, authors referred to significant increase of signal (figure 5a) but no SD shown in the figure. Please add SD in the figure.

Also, in the same paragraph (line 272), authors said that “significant at the last hour of the assay…” but if we observe the figure 5a we see that for 8 ng/mL CIP the significant difference from control was observed the last 20 min.

Round 2

Reviewer 2 Report

Dear Authors, Dear Editor,

It seems that, in spite of my recommendation (to reject the manuscript), manuscript 550739 will be published in Sensors. There is not much I can do but to accept this outcome of the evaluation process.

The authors could still correct the following two issues:

1.) As mentioned in my review of the original manuscript, manuscript 550739 lacks significant novelty when compared (first of all) to Cevenini, L.; Calabretta, M. M.; Tarantino, G.; Michelini, E.; Roda, A. Smartphone-Interfaced 3D Printed Toxicity Biosensor Integrating Bioluminescent “Sentinel Cells.” Sens. Actuators B Chem. 2016, 225, 249–257. Therefore, citing this paper is a must.

2.) As also mentioned in my review of the original manuscript, milk components reduce the bioavailability of ciprofloxacin (CIP). Therefore, the system described by the authors in their manuscript might underestimate the actual concentration of this antibiotic in milk. In yet other words, if the developed system “sees” 8 ng/mL of CIP in milk, the actual concentration might be larger than 8 ng/mL (not lower, as mentioned by the authors both in their response to the reviewer and in their manuscript). Please correct this mistake.

There is no need to return this manuscript for yet another round of evaluation. I trust the authors that they will do their best to correct these two small issues.

Author Response

We thank the reviewer for the detailed examination of the revision. The suggested modification has been done accordingly. The changes have been marked by “tracked changes” in the revised manuscript. The point-by-point response is as follows:

Q1: As mentioned in my review of the original manuscript, manuscript 550739 lacks significant novelty when compared (first of all) to Cevenini, L.; Calabretta, M. M.; Tarantino, G.; Michelini, E.; Roda, A. Smartphone-Interfaced 3D Printed Toxicity Biosensor Integrating Bioluminescent “Sentinel Cells.” Sens. Actuators B Chem. 2016, 225, 249–257. Therefore, citing this paper is a must.

A1: The reference has been added as no. 24 in the revised manuscript.

Q2.) As also mentioned in my review of the original manuscript, milk components reduce the bioavailability of ciprofloxacin (CIP). Therefore, the system described by the authors in their manuscript might underestimate the actual concentration of this antibiotic in milk. In yet other words, if the developed system “sees” 8 ng/mL of CIP in milk, the actual concentration might be larger than 8 ng/mL (not lower, as mentioned by the authors both in their response to the reviewer and in their manuscript). Please correct this mistake.

A2: To make the description more clear, we modify the sentence to :”… the LCS “sees” as 8 ng/mL may actually be lower than the initial concentration.”.